# Analysis of the Volatile Components in *Selaginella doederleinii* by Headspace Solid Phase Microextraction-Gas Chromatography-Mass Spectrometry

**DOI:** 10.3390/molecules25010115

**Published:** 2019-12-27

**Authors:** Xian-kui Ma, Xiao-fei Li, Jian-yong Zhang, Jie Lei, Wei-wei Li, Gang Wang

**Affiliations:** 1School of Pharmacy, Zunyi Medical University, Zunyi, Guizhou 563003, China; 2Key Laboratory of Basic Pharmacology of Ministry of Education and Joint International Research Laboratory of Ethnomedicine of Ministry of Education, School of Pharmacy, Zunyi Medical University, Zunyi, Guizhou 563003, China

**Keywords:** *Selaginella doederleinii*, HS-SPME, GC-MS, volatile profile

## Abstract

*Selaginella doederleinii* (SD) is a perennial medicinal herb widely distributed in China. In this study, the volatile components of SD from two regions (24 batches), namely Zhejiang and Guizhou, were determined by combining headspace solid phase microextraction and gas chromatography-mass spectrometry (HS-SPME/GC-MS). After investigating different influence factors, the optimal conditions for extraction were as follows: The sample amount of 1 g, the polydimethylsiloxane-divinylbenzene (PDMS-DVB) fiber of 65 µm, the extraction time of 20 min, and the extraction temperature of 100 °C. Based on the above optimum conditions, 58 volatiles compounds, including 20 terpenes, 11 alkanes, 3 alcohols, 6 ketones, 3 esters, 11 aldehydes, 1 ether, 1 aromatic, 1 phenol, and 1 furan, were found and identified in SD. Furthermore, hierarchical cluster analysis and principal component analysis were successfully applied to distinguish the chemical constituents of SD from two regions. Additionally, anethol, zingerone, 2,4-di-tert-butylphenol, ledene, hexyl hexanoate, α-cadinol, phytone, hinesol, decanal, octadecene, cedren, 7-tetradecene, copaene, β-humulene, 2-butyl-2-octenal, tetradecane, cedrol, calacorene, 6-dodecanone, β-caryophyllene, 4-oxoisophorone, γ-nonanolactone, 2-pentylfuran, 1,2-epoxyhexadecane, carvacrol, n-pentadecane, diisobutyl phthalate, farnesene, n-heptadecane, linalool, 1-octen-3-ol, phytane, and β-asarone were selected as the potential markers for discriminating SD from 24 habitats in Zhejiang and Guizhou by partial least squares discrimination analysis (PLS-DA). This study revealed the differences in the components of SD from different regions, which could provide a reference for the future quality evaluation.

## 1. Introduction

*Selaginella doederleinii* (SD) is widely distributed in China and used as a traditional Chinese medicine [1]. Clinically, it is often applied to treat cancer such as nasopharyngeal carcinoma and esophageal cancer. The main components in SD include flavonoids, alkaloids, lignans, organic acids, and volatiles [2]. These components have been shown to have a wide range of activities. For instance: (a) Amentoflavone, robustaflavone, heveaflavone, and 7, 4′, 7′′, 4′′′-tetra-O-methyl- amentoflavone possessed an anti-proliferation effect on human cancer cells [3]; (b) nine kinds of biflavones showed DPPH free radical scavenging activities [4]; (c) terpenoids were reported to have antibacterial activity [5]. Due to the extensive activities, SD is becoming a more and more important medicinal plant.

The combination of headspace solid phase microextraction and gas chromatography-mass spectrometry (HS-SPME/GC-MS) is an important method to study the volatiles [6]. Compared with the traditional method, HS-SPME uses fused silica fibers coated with a stationary phase to achieve direct adsorption of volatile components [7]. The benefits of HS-SPME-GC/MS for SD analysis includes: (1) The technique integrates sampling, extraction, and analysis, and thus it is easy to operate and shortens the analysis time; (2) it has the advantages of low sample consumption and high sensitivity [8]. Due to the obvious advantage, HS-SPME is commonly coupled with GC-MS for the efficient, sensitive, and rapid analysis of the volatiles [9]. At the same time, based on the NIST database search, this technology can realize the discovery of trace volatile components in complex sample systems [10]. For example, Nie et al. used HS-SPME-GC/MS to analyze the volatile components of Sichuan Dark brick tea (*Camellia sinensis*) and Sichuan Fuzhuan brick tea. The results showed that there are 37 common volatile compounds in the above two teas [11]. Wang et al. analyzed the volatiles in *Polygonum multiflorum* by HS-SPME/GC-MS technique and the optimal extraction process by single factor analysis was determined. The characteristic compounds of *Polygonum multiflorum* included pulegone, cis-isopulegone, menthone [12]. However, HS-SPME/GC-MS has not been used to analyze the constituents of SD from different habitats.

Plants contain a large number of complex chemical components. Moreover, the chemical components of the same plant in different regions are different [13]. Chemometrics, a new branch of chemistry, integrates chemistry, statistics, and computer science to obtain large amounts of chemical information and clarify the complex relationship between different species. Hierarchical cluster analysis (HCA), principal component analysis (PCA), and partial least squares discriminant analysis (PLS-DA) are the representative methods to analyze chemical data, establish a model for quality control, and perform classification. Recently, multivariate statistical analysis was usually integrated with chemical analysis, which played an important role for the quality evaluation of natural products [14,15,16,17]. The statistical regularity of multiple objects and multiple indicators could be analyzed, which is helpful for component-based classification [18]. The combination of various methods in different fields to study chemical components of plants has become a trend [19].

According to preliminary studies, we found the volatile oils of SD had antioxidant and antitumor activity [20,21] and tried to explore chemical components in the volatiles on this basis. Herein, in this work, a comprehensive strategy that integrated HS-SPME/GC-MS and multivariate statistical analysis was proposed for the first time to compare the chemical constituents of SD from different habitats, and to identify and classify SD from multiple angles. HS-SPME/GC-MS was applied to analyze the chemical constituents of SD from different habitats. Multivariate statistical methods such as HCA and PCA were performed to classify SD from Zhejiang and SD from Guizhou. Furthermore, PLS-DA was used to find the potential chemical markers for the classification of SDs from the two regions. This study is helpful to improve the method for quality control and increase the knowledge of components in SD.

## 2. Results and Discussion

### 2.1. Optimization of the Parameters for HS-SPME

The different factors, including different fibers (PDMS, PDMS/CAR, DVB/CAR/PDMS, and PDMS/DVB), sample amount (0.8, 0.9, 1.0, 1.1, and 1.2 g), extraction time (5, 10, 15, 20, and 25 min), and temperature (80, 90, 100, 110, and 120 °C), were investigated and optimized to extract volatile components from *S. doederleinii*. The detailed discussion was as follows.

Different types of fibers have different adsorption and retention capacities and are suitable for different analytes [22]. In this paper, four fibers (PDMS, CAR/PDMS, DVB/CAR/PDMS, and PDMS/DVB) were applied to extract the volatiles from *S. doederleinii*. In gas chromatography, the sample extracted by PDMS/DVB fiber showed the largest peak area. As shown in Figure 1A, the total number of peaks obtained by PDMS/DVB fiber was also more than that obtained by the other three fibers, which proved that PDMS/DVB fiber had good adsorption and retention capacities for the extraction of volatile components from *S. doederleinii*. Therefore, PDMS/DVB was identified as the optimal fiber in this study.

Temperature is an important parameter for the SPME extraction process. Since extraction by this method is an exothermic process, it affects the diffusion rate of the analytes into the fiber coating [23]. One gram samples were extracted with 65 μm PDMS/DVB fiber for 20 min. Volatile components were detected at 80, 90, 100, 110, and 120 °C, respectively. Each sample was repeated three times. As can be seen in Figure 1B, when the temperature reached 100 °C, the total peak area and the number of the volatile oil were the highest. Therefore, 100 °C was the optimal extraction temperature.

The time that the fiber is in contact with the sample has a significant effect on the chromatographic peak areas of the volatile oils [24]. The extraction time was optimized under the conditions of 65 μm PDMS/DVB fiber, 1.0 g sample amount, and 60 °C, and the total peak areas of the volatiles were analyzed at 5, 10, 15, 20 and 25 min (see Figure 1C). The results showed that the total peak area and the number of the volatile oil were the highest at 20 min. Therefore, the optimal extraction time was set to 20 min.

After screening the effects of PDMS/DVB fiber, extraction temperature, and time, the influence of sample amount on the extraction efficiency of the volatile oils is shown in Figure 1D. The extraction time was 20 min, and the extraction temperature was set at 100 °C. Under these conditions, different sample amounts were analyzed and optimized. As shown in Figure 1D, the sample amount of 1.0 g had a significant effect on the extraction efficiency of volatile oil. Therefore, it was appropriate to select 1.0 g as the sample amount.

### 2.2. Fingerprints of Selaginella doederleinii

Under the optimum extraction conditions, HS-SPME-GC-MS was used to identify the volatile oil of SD samples from Guizhou and Zhejiang provinces. The experiment was repeated three times. The total peak area, RI, and relative percentage content of the volatile oils from the herbs were calculated [25]. The total ion chromatogram (TIC) of SD from different origins are shown in Figure 2. As shown in Table 1 and Table 2, a total of 58 volatile compounds, including 20 terpenes, 11 alkanes, 3 alcohols, 6 ketones, 3 esters, 11 aldehydes, 1 ethers, 1 aromatic, 1 phenol, and 1 furan were found and identified in SD. Among them, 45 volatile components were reported in the volatile oils of SD for the first time.

The concentration of each component differed among the volatile oils from 24 habitats of SD. For instance, the concentration of anethol in the volatiles of SD from Guizhou was over 20 times higher than that in the volatile oils of SD from Zhejiang. Consequently, in order to find the marker compounds for the comparison of SD Guizhou and SD Zhejiang, an unpaired T-test was employed for analyzing the relative contents of the common peaks of SD [26]. As shown in Figure 2 and Table 1, there were significant differences in volatile components between them. The main components of SD from Guizhou included: Anethol (12.78 ± 0.31%), cedrol (6.97 ± 0.18%), 2,4-nonadienal (5.60 ± 0.34%), 2-pentylfuran (5.48 ± 0.13%), phytone (5.44 ± 0.24%), hexadecane (3.19 ± 0.39%), linalool (3.36 ± 0.23%), and aromadendrene (3.70 ± 0.15%). The main components of SD from Zhejiang included: Phytone (11.41 ± 1.21%), 2-pentylfuran (7.50 ± 0.52%), carvacrol (2.21 ± 0.48%), ledene (5.48 ± 0.13%), cedrol (4.66 ± 0.79%), 2,4-nonadienal (4.65 ± 0.76%), aromadendrene (4.80 ± 0.56%), β-caryophyllene (2.54 ± 0.19%), zingerone (5.84 ± 0.59%), and n-hexadecane (3.61 ± 0.43%). Among the above presented components, the relative contents of phytone, cedrol, aromadendrene, hexadecane, and 2-pentylfuran were more than 3% in all SD samples. In general, the contents of aldehydes, alcohols, alkanes, ethers, and aromatics in SD from Guizhou Province were higher than those from Zhejiang Province (Table 2). It was worth noting that the difference among aldehydes was particularly significant. The aldehydes in Zhejiang and Guizhou accounted for 11.27 ± 0.55% and 16.30 ± 0.57%, respectively, including 2-heptenal, 2-octenal, 1-nonanal, 2-nonenal, decanal, 2,4-nonadienal, citral, 2,4-decadienal, dodecanal, 2-butyl-2-octenal, and tetradecanal. There was a significant difference in the contents of aldehydes and alkanes between the two regions (*p* < 0.05). However, the contents of ketones, furans, phenols, terpenes, and esters in SD from Guizhou Province were lower than those from Zhejiang Province, among which the difference in ketones was noticeable. The ketones in Zhejiang and Guizhou accounted for 14.93 ± 0.68% and 11.61 ± 0.46%, respectively, including 3-octanone, 4-oxoisophorone, 6-undecanone, 6-dodecanone, 4-(4-methoxyphenyl)-2-butanone, and phytone.

Whether in SD Guizhou or SD Zhengjiang, terpenes were the highest percentage compounds and accounted for nearly half of all oil samples, which represented the chemical characteristic of SD well. The most abundant terpenes (Table 1) in SD, of which the concentrations were above 1%, were linalool (2.37–3.36%), aromadendrene (3.70–4.80%), zingerone (1.51–5.48%), cedrol (4.66–6.97%), anethol (0.56–12.78%), and α-cadinol (1.15–2.21%). The terpenes were generally considered to be active constituents in natural products [27], which exhibited antibacterial, antitumor, anti-wrinkle, antioxidative, anti-tussive, analgesia, and immune-regulatory effects [28]. For example, Gunaseelan [29] and others found the treatment of linalool prevented acute ultraviolet-B-induced hyperplasia, lipid peroxidation, and anti-oxidation loss of mice skin and could further inhibited overexpression of cyclooxygenase-2 and ornithine decarboxylase in mice skin. Zhang [30] and others first revealed that cedrol improved the level of 5-hydroxytryptamine (5-HT), decreased the ratio of 5-hydroxyindoleacetic acid/5-HT, and had significant anxiolytic effect on female mice via the 5-HT pathways.

However, the pharmacological activity of terpene from SD has not been reported. In our preliminary experiment, the volatile oils have been observed to inhibit proliferation of tumor cell and eliminate free radicals. Further study is needed to explore the bioactivities of the terpene of the volatile oils.

### 2.3. PCA

To provide more information about differentiation of the origins of SD, PCA was performed based on the 58 common peak areas as new derived variables (principal components, PCs) with eigen values >1, which could adequately summarize the original information. The first two PCs explained approximately 81.6% of the original data variability and the predictive ability of the model (Q2) was 53.5%, which indicated that it was a good model (Q2 > 0.50). According to the PCA score plot model as shown in Figure 3A, one group was the Zhejiang sample point (ZJ1~ZJ11, positive position) and the other was Guizhou sample point (GZ1~GZ13, negative position), which showed that both domains (Zhejiang SD and Guizhou SD) were obviously separated from each other. Moreover, the dots that presented GZ1~GZ13 were close to each other, suggesting the 13 batches of SD from Guizhou were similar. The dots of ZJ1~ZJ11 were relatively scattered, indicating diversification of the 11 batches of Zhejiang SD. It was indicated that the chemical composition of Zhejiang SD was not as stable as Guizhou SD. As shown in the PCA loading plot model (in Figure 3B), the first principal component (PC1) explained 57.2% of the variance and was mainly represented by the linear combination of the following variables: Zingerone (49), 7-tetradecene (26), dodecanal (20), trans,trans-2,4-nonadienal (14), tetradecane (27), beta-asarone (44), hexyl hexanoate (25), copaene (28), 2-butyl-2-octenal (24) and trans-2-decenal (10). PC2 identified 24.4% of the variance and was mainly expressed by a linear combination of the variables, including 2,4-di-tert-butylphenol (40), 4-allylanisole (12), carvacrol (21), citral (15), 1,2-epoxyhexadecane (55), 2,6-di-tert-butyl-p-benzoquinone (35), and 1-nonanal (7).

### 2.4. HCA

HCA is an analysis process that groups a collection of physical or abstract objects into multiple classes composed of similar objects [38]. It has developed rapidly in the past 20 years and has been widely used in the fields of commercial pattern recognition, plant classification, etc. [39]. As shown in Figure 4, the results of clustering analysis indicated that the 24 batches of SD samples were mainly divided into two categories: The first type was Zhejiang SD (ZJ-1 to ZJ-11); the second type was Guizhou SD (GZ-1 to GZ-13). There were some differences in the components of SD from various habitats. It might be due to some factors such as soil, climate, and harvesting season. It could be seen that the samples were separated by clustering analysis, and the differences among the samples were accurately described.

### 2.5. PLS-DA

After HCA and PCA analysis of Zhejiang SD and Guizhou SD, partial least squares discrimination analysis (PLS-DA) was applied to visualize the variations among SD samples. According to the PLS-DA model as shown in Figure 5, the cross-validated predictive capability (Q2), variance (R2Y), and variance (R2X) of the model are 0.988, 0.995, and 0.921, respectively, indicating that the model was stable. Based on the PLS-DA, a loadings plot was drawn to exhibit the contribution of each variable to the discrimination [40]. Variable important plot (VIP) score is an indicator of the correlation of variables. When the volatile oil has a VIP value greater than 1, it indicates that they have an above-average effect [41] and can be used to distinguish *S. doederleinii* in Zhejiang and Guizhou, namely anethol(19), zingerone(49), 2,4-di-tert-butylphenol(40), ledene(42), hexyl hexanoate(25), α-cadinol(52), phytone(58), hinesol(48), decanal(13), octadecene(56), cedren(30), 7-tetradecene(26), copaene(28), beta-humulene(37), 2-butyl-2-octenal(24), tetradecane(27), cedrol(47), calacorene(43), 6-dodecanone(23), β-caryophyllene(31), 4-oxoisophorone(9), γ-nonanolactone(22), 2-pentylfuran(5), 1,2-epoxyhexadecane(55), carvacrol(21), n-pentadecane(38), diisobutyl phthalate, (53) farnesene(29), n-heptadecane(54), linalool(8), 1-octen-3-ol(2), phytane(57), and β-asarone(44). Therefore, these substances were important markers for the identification and quality control of *S. doederleinii*. Considering that the volatile oil of *S. doederleinii* exhibited antioxygenic and antitumor effects, it is worth studying whether the above compounds are responsible for these activities.

Based on the above chemometric analysis, the content of the volatile compounds showed great differences between different habitats of SD. It is deduced that the difference on SD might be related to the harvest season of the plant, the geographic regions, the processing technology of the herbal medicines, etc. [42].

## 3. Material and Methods

### 3.1. Materials

A total of 11 batches of Zhejiang SD (batch number ZJ01, ZJ02, ZJ03, ZJ04, ZJ05, ZJ06, ZJ07, ZJ08, ZJ09, ZJ10, and ZJ11) and 13 batches of Guizhou SD (batch number GZ01, GZ02, GZ03, GZ04, GZ05, GZ06, GZ07, GZ08, GZ09, GZ10, GZ11, GZ12, and GZ13) were purchased from Hangzhou Chinese Herbal Medicine Trading Center (Zhejiang Province) and Guiyang Chinese Herbal Medicine Market (Guizhou Province). These samples were authenticated by Zhang Yujin (an associate professor from Zunyi Medical University) and deposited in the chemical laboratory. n-Hexane of analytical grade was purchased from Suzhou Boyuan Chemical Co., LTD (Suzhou, China). n-Alkane (C_8_–C_30_) series were provided by Sa Lan Chemical Co., LTD (Shanghai, China). SPME holder and 100 µm Polydimethylsiloxane (PDMS), 75 µm Carboxen/Polydimethylsiloxane (CAR/PDMS), 65 µm Polydimethylsiloxane/Divinylbenzene PDMS/DVB, and 50/30 µm Divinylbenzene/Carboxen/ Polydimethylsiloxane (DVB/CAR/PDMS) fiber were purchased from Supelco Inc (Bellefonte, PA, USA). HS-SPME unit and transparent sample bottle were purchased from Kanglin Technology Co., LTD (Beijing, China).

### 3.2. Sample Preparation

The purchased herbs were cut, washed and dried naturally at room temperature. Then, the raw material of SD was crushed at a Chinese herbal medicine crusher (WK-1000A, Ruiqianshun Co. Ltd., Chongqing, China). According to a previous report [43], particle size was controlled through the 100 mesh sieve (36.358–95.719 μm), which was measured by a winner-2006 laser particle diameter analyzer (Weina technology Co. Ltd., Qingdao, China).

### 3.3. Sample Pretreatment

To eliminate the residue on the fiber coating, the four fibers were treated in a GC inlet at 250 °C for 5 min. A certain amount of the medicinal material was pulverized and passed through a 100 mesh sieve. The medicinal material powder was placed in a 10 mL glass vial, which was covered with a polytetrafluoroethylene (PTFE) septum and an aluminum lid. The needle was inserted into the vials. Different fibers were exposed to the headspace above the sample at a certain temperature to extract the medicinal material for a period of time. Next, the needle was transferred to the GC-MS inlet. After 2 min of thermal desorption of the fibers, the volatile oils were measured and analyzed by GC-MS.

### 3.4. GC-MS Analysis

An Agilent Technologies 6890 GC system (Agilent Technologies Inc., Palo Alto, CA, USA) coupled with an Agilent Technologies 5973 mass spectrometer (Agilent Technologies Inc., Palo Alto, CA, USA) was applied for volatility component analysis (Appendix A). A HP-5MS capillary column (30m × 0.25mm × 0.25μm) was used to separate the volatiles. The temperature procedure was as following: 0–5 min, 50–50 °C; 5–20 min, 50–200 °C; 20–23 min, 200–200 °C; 23–27 min, 200–250 °C; 27–32 min, 250–250 °C. The splitless mode was adopted. High-purity helium was applied as the carrier gas, and the rate was 1 mL/min. The working conditions of MS were as follows: The electron ionization energy was 70 eV, the full-scan acquisition was used in the range of 10–800 amu, the ion source temperature was 230 °C, the emission current was 200 μA, and solvent delay was set at 5 min.

Retention indices (RI) of each volatile, a parameter for qualitative indicators of GC, were calculated using the retention value of two adjacent n-alkane standards. The preliminary identification of the volatiles was done by comparing the mass spectra with National Institute of Standards and Technology (NIST) 14.0 MS database and Wiley National Institute of Standards and Technology (Wiley NIST) 14.0 MS database. These volatiles were further confirmed by matching the reference literature records and retention time of authentic standards, as well as contrasting the calculated RI with that which was recorded in the NIST network database (http://webbook.nist.gov/chemistry/) [44]. Identified volatile oils were semi-quantified by comparing the peak areas of each volatile with total peak areas. Relative percentage content of the volatile compounds was calculated via the peak area normalization method using the following formula:Mi% = Ai/∑Ai *×* 100%(1)
where Mi is the percentage content of the measured volatile i; ∑Ai is the total peak area; Ai is the peak area of the measured volatile.

### 3.5. Data Analysis

All experiments were carried out three times in parallel. Statistical significance (*p* < 0.05) was analyzed by one-way ANOVA analysis of SPSS 23.0 (SPSS Inc., Chicago, IL, USA). Principal component analysis (PCA) and partial least squares-discrimination analysis (PLS-DA) were performed by SIMCA P14 (Umetrics, Umea, Sweden). Hierarchical cluster analysis was proposed by SPSS 23.0 software (SPSS Inc., Chicago, IL, USA).

## 4. Concluding Remarks

The HS-SPME-GC-MS technique was applied to study the differences in the volatile oils of *S. doederleinii* from different habitats. The results showed that the optimization of the significant factors affecting sorption process such as different fibers, extraction temperature, sample amount, and extraction time, was completed by a single factor experiment design. The results showed that the PDMS/DVB fiber of 65 µm was most suitable for the isolation of the volatiles from SD. The other optimum conditions were as follows: Sample amount of 1.0 g, extraction time of 20 min, and extraction temperature of 100 °C, respectively.

Based on the optimal conditions, there were common 58 volatile substances in Guizhou SD and Zhejiang SD. Terpenes were found to comprise the largest chemical class in SD. Phytone (11.41%) and cis-anethol (12.78%) were found to be the most abundant volatile component in SD from Guizhou and Zhengjiang, respectively.

Finally, chemometric analysis indicated that the difference of SD in two provinces was very significant. The acquired data set was submitted to PCA and HCA, and the corresponding Guizhou SD and Zhejiang SD discrimination pattern according to 24 habitats was successfully established. Moreover, based on SLDA analysis, 34 volatile constituents (VIP > 1) were important markers for the differentiation between Guizhou SD and Zhejiang SD, and quality control of SD.

In this study, it can be concluded that the producing area has an obvious influence on the contents of volatile oils of SD. Attention should be paid to the chemical differences among different habitats of SD, and their main components such as terpenes. The biological activity of components in SD needs to be further revealed. The results in this work provides a basis for the quality evaluation of SD in the future.

## Figures and Tables

**Figure 1 molecules-25-00115-f001:**
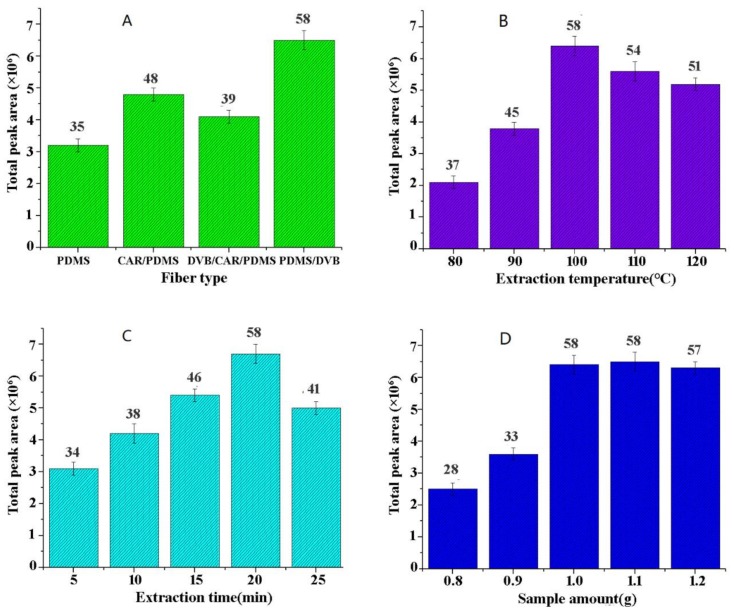
Effects of extraction parameters on the volatiles of *Selaginella doederleinii*. (**A**) solid phase microextraction (SPME) fiber; (**B**) extraction temperature; (**C**) extraction time; (**D**) sample amount.

**Figure 2 molecules-25-00115-f002:**
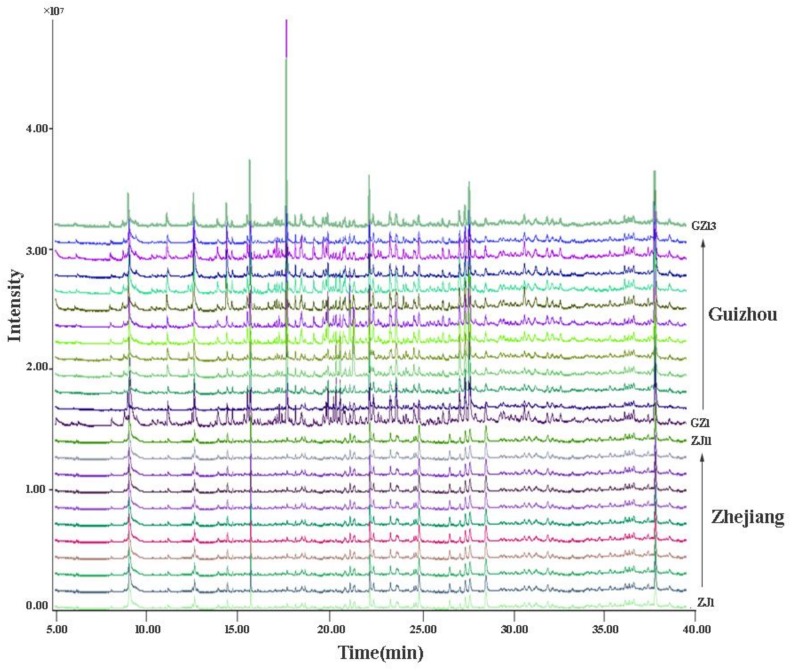
Typical chromatogram of the volatile components from *Selaginella doederleinii* by headspace solid phase microextraction and gas chromatography-mass spectrometry (HS-SPME/ GC–MS).

**Figure 3 molecules-25-00115-f003:**
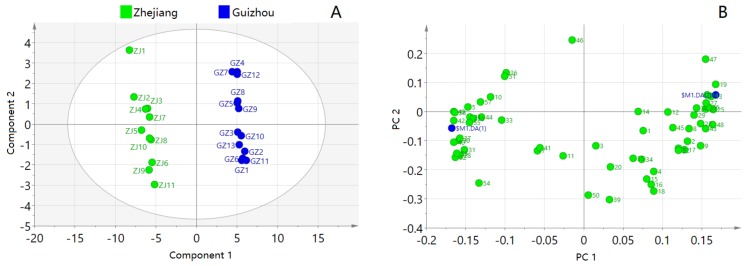
Score plot (**A**) and loading plot (**B**) of principal component analysis of 24 samples of *Selaginella doederleinii* from two districts.

**Figure 4 molecules-25-00115-f004:**
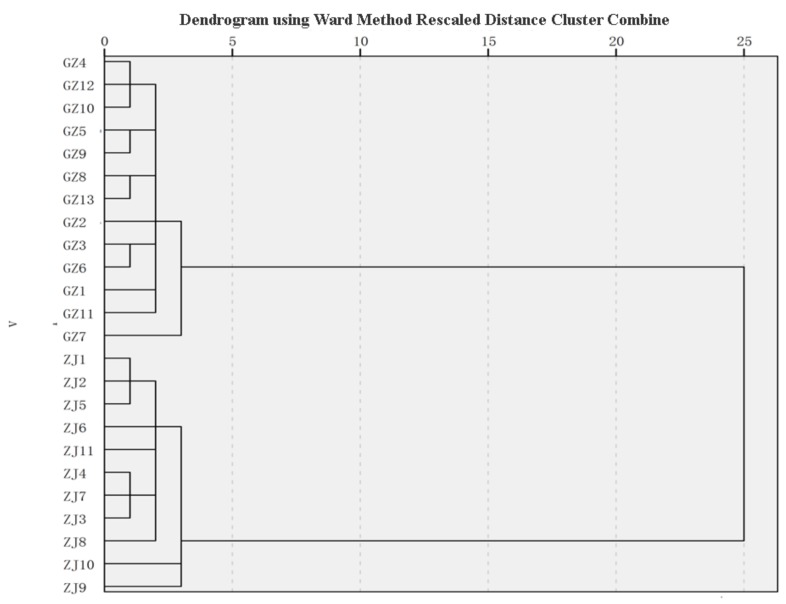
Hierarchical cluster analysis (HCA) of 24 samples of *Selaginella doederleinii* from two districts using Ward’s method based on the Euclidean distance.

**Figure 5 molecules-25-00115-f005:**
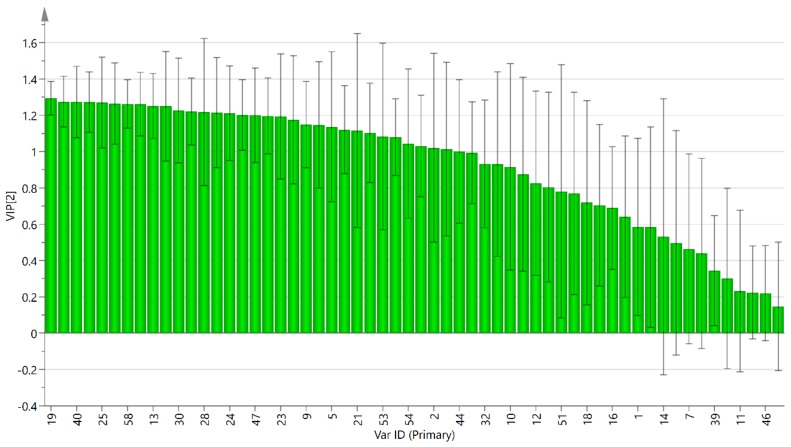
Variable important plot (VIP) of identified compounds from 24 samples of *Selaginella doederleinii* from two districts based on partial least squares discrimination analysis (PLS-DA).

**Table 1 molecules-25-00115-t001:** Relative contents (%) of volatile oil of *Selaginella doederleinii* by HS-SPME/GC-MS.

No	RI^a^	RI^b^	Rt/Min	Identification	Compounds	C	CAS	Peak Areas Ratios (%)		p Value	Vip
								Zhejiang (n = 11)	Guizhou (n = 13)		
1	951	956	8.188	STD,MS,RI	2-heptenal	A	18829-55-5	0.35 ± 0.03	0.67 ± 0.02	0.029	0.593
2	979	984	8.874	STD,MS,LR,RI	1-octen-3-ol	B	3391-86-4	0.21 ± 0.02	0.82 ± 0.12	0.006	1.033
3	983	989	8.992	MS,LR,RI	3-octanone	C	106-68-3	0.33 ± 0.16	0.37 ± 0.09	0.679	0.151
4	986	967	9.095	MS,LR,RI	1-heptanol*	B	111-70-6	0.17 ± 0.05	0.41 ± 0.16	0.009	0.714
5	989	988	9.187	STD,MS,RI	2-pentylfuran	D	3777-69-3	7.50 ± 1.25	5.48 ± 0.13	0.003	1.150
6	1059	1056	11.245	STD,MS,LR,RI	2-octenal*	A	2548-87-0	1.11 ± 0.23	1.92 ± 0.38	0.003	1.226
7	1103	1102	12.579	STD,MS,LR,RI	1-nonanal	A	124-19-6	0.75 ± 0.31	0.59 ± 0.25	0.099	0.470
8	1104	1106	12.693	STD,MS,LR,RI	linalool	H	78-70-6	2.37 ± 0.68	3.36 ± 0.23	0.012	1.044
9	1143	1142	13.978	STD,MS,LR,RI	4-oxoisophorone*	C	1125-21-9	0.76 ± 0.31	1.45 ± 0.11	0.017	1.163
10	1154	1159	14.475	MS,LR,RI	2-nonenal	A	2463-53-8	2.45 ± 0.16	1.46 ± 0.06	0.009	0.927
11	1183	1181	14.767	MS,LR,RI	terpinen-4-ol*	H	562-74-3	0.85 ± 0.26	0.83 ± 0.05	0.475	0.237
12	1204	1201	15.405	STD,MS,RI	4-allylanisole*	F	140-67-0	0.31 ± 0.18	0.73 ± 0.12	0.001	0.836
13	1207	1203	15.594	STD,MS,LR,RI	decanal*	A	112-31-2	0.15 ± 0.06	2.05 ± 0.38	0.008	1.267
14	1216	1214	15.739	STD,MS,LR,RI	2,4-nonadienal*	A	5910-87-2	4.65 ± 0.76	5.6 ± 0.34	0.050	0.537
15	1247	1241	17.092	STD,MS,LR,RI	citral	A	5392-40-5	0.37 ± 0.08	0.55 ± 0.16	0.020	0.651
16	1257	1243	17.203	MS,LR,RI	2,4-decadienal	A	2363-88-4	0.21 ± 0.05	0.43 ± 0.35	0.012	0.699
17	1259	1255	17.322	STD,MS,LR,RI	6-undecanone*	C	927-49-1	0.17 ± 0.09	0.79 ± 0.18	0.007	1.005
18	1260	1254	17.468	MS,RI	2-decen-1-ol*	B	22104-80-9	0.15 ± 0.07	0.38 ± 0.17	0.008	0.730
19	1267	1262	17.727	STD,MS,RI	anethol*	H	104-46-1	0.56 ± 0.15	12.78 ± 0.3	0.003	1.310
20	1276	Nr	18.208	STD,MS,LR,RI	dodecanal*	A	112-54-9	0.74 ± 0.16	0.79 ± 0.13	0.346	0.307
21	1290	1292	18.543	STD,MS,RI	carvacrol*	H	499-75-2	2.21 ± 0.48	1.14 ± 0.24	0.011	1.129
22	1356	1355	19.721	STD,MS,RI	γ-nonanolactone*	J	104-61-0	0.13 ± 0.03	0.69 ± 0.05	0.015	1.161
23	1358	1355	19.883	MS,RI	6-dodecanone*	C	6064-27-3	0.35 ± 0.09	1.21 ± 0.31	0.009	1.208
24	1375	1378	19.980	MS,LR,RI	2-butyl-2-octenal*	A	13019-16-4	0.23 ± 0.04	1.91 ± 0.58	0.007	1.226
25	1379	1385	20.288	STD,MS,LR,RI	hexyl hexanoate*	J	6378-65-0	0.37 ± 0.04	1.28 ± 0.18	0.010	1.285
26	1394	1396	20.423	MS,RI	7-tetradecene*	E	10374-74-0	0.16 ± 0.03	2.16 ± 0.42	0.021	1.236
27	1403	1400	20.644	STD,MS,LR,RI	tetradecane	E	629-59-4	0.15 ± 0.02	1.77 ± 0.16	0.005	1.216
28	1415	1417	20.817	MS,LR,RI	copaene*	H	3856-25-5	1.62 ± 0.12	0.8 ± 0.25	0.017	1.233
29	1446	1461	20.871	MS,RI	farnesene*	H	502-61-4	0.14 ± 0.05	1.21 ± 0.31	0.013	1.093
30	1447	1432	21.185	MS,RI	cedren*	H	469-61-4	1.80 ± 0.17	0.65 ± 0.03	0.019	1.242
31	1448	1437	21.379	STD,MS,LR,RI	β-caryophyllene	H	87-44-5	2.54 ± 0.19	1.3 ± 0.57	0.008	1.190
32	1450	1440	21.752	MS,RI	calarene*	H	17334-55-3	0.13 ± 0.06	0.53 ± 0.09	0.022	0.944
33	1452	1446	22.238	MS,LR,RI	aromadendrene	H	489-39-4	4.80 ± 0.56	3.7 ± 0.15	0.001	0.814
34	1471	1473	22.454	STD,MS,RI	4-(4-methoxyphenyl)-2-butanone*	C	104-20-1	1.91 ± 0.31	2.15 ± 0.33	0.034	0.591
35	1472	1472	22.731	MS,RI	2,6-di-tert-butylbenzoquinone*	I	719-22-2	0.37 ± 0.08	0.43 ± 0.08	0.077	0.504
36	1489	1488	23.367	STD,MS,RI	β-ionone	H	14901-07-6	2.35 ± 0.11	1.37 ± 0.37	0.043	0.779
37	1492	1497	23.432	MS,RI	β-humulene*	H	116-04-1	2.05 ± 0.13	0.68 ± 0.19	0.018	1.230
38	1497	1500	23.697	STD,MS,LR,RI	n-pentadecane	E	629-62-9	1.02 ± 0.05	2.8 ± 0.15	0.019	1.117
39	1502	Nr	24.102	STD,MS,LR,RI	tetradecanal*	A	124-25-4	0.26 ± 0.06	0.33 ± 0.32	0.365	0.353
40	1507	1513	24.242	STD,MS,LR,RI	2,4-di-tert-butylphenol*	G	96-76-4	1.38 ± 0.11	0.42 ± 0.08	0.023	1.289
41	1515	1519	24.674	MS,LR,RI	β-bisabolene*	H	495-61-4	1.54 ± 0.16	1.42 ± 0.45	0.119	0.445
42	1532	1534	24.934	MS,RI	ledene*	H	21747-46-6	5.48 ± 0.34	1.5 ± 0.28	0.027	1.288
43	1560	1562	26.252	MS,RI	calacorene*	E	21391-99-1	0.16 ± 0.03	0.87 ± 0.36	0.031	1.211
44	1592	1596	26.597	STD,MS,LR,RI	β-asarone*	H	5273-86-9	1.81 ± 0.12	0.78 ± 0.18	0.016	1.013
45	1596	1593	27.165	STD,MS,LR,RI	1-hexadecene*	E	629-73-2	1.42 ± 0.13	2.26 ± 0.25	0.056	0.885
46	1605	1600	27.467	STD,MS,LR,RI	n-hexadecane	E	544-76-3	3.61 ± 0.43	3.19 ± 0.39	0.690	0.228
47	1609	1606	27.710	STD,MS,LR,RI	cedrol	H	77-53-2	4.66 ± 0.79	6.97 ± 0.18	0.009	1.216
48	1611	1632	28.018	MS,LR,RI	hinesol*	H	23811-08-7	0.36 ± 0.06	0.98 ± 0.56	0.025	1.277
49	1639	1645	28.575	STD,MS, RI	zingerone*	H	122-48-5	5.84 ± 0.59	1.51 ± 0.41	0.012	1.291
50	1644	Nr	29.406	STD,MS, RI	1-chlorooctadecane*	E	3386-33-2	0.55 ± 0.12	0.72 ± 0.21	0.879	0.790
51	1643	1649	30.698	MS,LR,RI	t-muurolol*	H	19912-62-0	1.68 ± 0.45	1.11 ± 0.15	0.002	0.234
52	1664	1666	30.914	MS,LR,RI	α-cadinol*	H	481-34-5	2.21 ± 0.16	1.15 ± 0.17	0.013	1.281
53	1691	Nr	31.940	STD,MS,RI	diisobutyl phthalate*	J	84-69-5	2.79 ± 0.34	0.82 ± 0.46	0.028	1.097
54	1701	1700	32.221	STD,MS,LR,RI	n-heptadecane	E	629-78-7	1.15 ± 0.15	0.81 ± 0.09	0.024	1.059
55	1706	1708	32.669	MS, RI	1,2-epoxyhexadecane*	E	7320-37-8	1.27 ± 0.16	0.51 ± 0.04	0.035	1.134
56	1799	1800	36.164	STD,MS,LR,RI	1-octadecene	E	112-88-9	1.27 ± 0.47	0.48 ± 0.35	0.016	1.266
57	1809	1810	36.641	MS,LR,RI	phytane*	E	638-36-8	1.89 ± 0.18	0.85 ± 0.31	0.008	1.026
58	1846	1847	37.812	MS,RI,LR	phytone	C	502-69-2	11.41 ± 1.2	5.44 ± 0.24	0.007	1.278

RI^a^—retention index calculated; RI^b^—retention index from the NIST network database, Nr—not recorded in database; *—compounds detected from *Selaginella doederleinii* for the first time; C—compound classification, A—aldehyde, B—alcohol, C—ketone, D—furan, E—alkane, F—ether, G—phenol, H—terpene, I—aromatic, J—ester; MS—confirmed by mass spectrum, RI—confirmed by retention index in database, STD—confirmed by reference standard, LR—confirmed by published literature [31,32,33,34,35,36,37]; P value—indicators for statistical significance among cultivars (*p* < 0.05); VIP—variable important plot.

**Table 2 molecules-25-00115-t002:** Different compound categories in *Selaginella doederleinii* from 24 habitats.

Compound Type	Numbers	Peak Areas Ratios (%)	
		Zhejiang (*n* = 11)	Guizhou(*n* = 13)
aldehyde	11	11.27 ± 0.55	16.30 ± 0.57
alcohol	3	0.53 ± 0.23	1.61 ± 0.13
ketone	6	14.93 ± 0.68	11.61 ± 0.46
furan	1	7.50 ± 0.52	5.48 ± 0.34
alkane	11	12.65 ± 1.12	16.42 ± 0.67
ether	1	0.31 ± 0.06	0.73 ± 0.16
phenol	1	1.38 ± 0.12	0.42 ± 0.07
terpene	20	45.00 ± 3.14	43.77 ± 2.16
aromatic	1	0.37 ± 0.08	0.43 ± 0.07
ester	3	3.29 ± 0.48	2.79 ± 0.24

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
