# Peer review of "Analysis of the Volatile Components in Selaginella doederleinii by Headspace Solid Phase Microextraction-Gas Chromatography-Mass Spectrometry"

_molecules, 2019, doi:10.3390/molecules25010115_

Round 1

Reviewer 1 Report

This article describes the application of using headspace solid-phase microextraction-gas chromatography-mass spectrometry to analyze headspace volatiles from Selaginella doederleinii (SD).  Samples were collected from two different geographical regions. Statistical analysis of the data collected from two regions has been performed to distinguish the chemical constituents of SD.  The authors showed that SD grown from different habitats has characteristic volatile components which may provide a way for quality evaluations.

The reviewer has specific comments below:

Page 1, line 36: it is not clear which chemical component in SD has been used for cancer treatment.

Page 2, line 44: authors state “In recent years, volatiles from SD have attracted wide attention [6]”, however, the cited reference was published in 1994, which was more than 20 years ago.

Page 2, line 44-46: authors state “The combination of headspace solid-phase microextraction and gas chromatography-mass spectrometry (HS-SPME/GC-MS) is an important method to study volatiles [7]”. The cited reference was not an HS-SPME/GC-MS work. Instead, it was microwave-assisted extraction work.

Page 2 line 51-52, some relevant literature below can be included in this paragraph:

M. Zakir Hossain, Barbara Bojko, Janusz Pawliszyn, Automated SPME–GC–MS monitoring of headspace metabolomic responses of E. coli to biologically active components extracted by the coating, Analytica Chimica Acta, Volume 776, 7 May 2013, Pages 41-49. Austin McDaniel, Lauren Perry, Qingzhong Liu, Wei-Chuan Shih, Jorn Yu, Toward the identification of marijuana varieties by headspace chemical forensics, Forensic Chemistry, Volume 11, December 2018, Pages 23-31.

Page 2, line 53-53, a typical SPME does not use the membrane adsorption technology. Can the author verify the statement here? The authors should cite the original design of the SPME from “J. Pawliszyn, Solid Phase Microextraction: Theory and Practice, Wiley-VCH, New York, 1997.”

Page 2, line 60, revise “Nie Cong-Ning et al” to “Nie et al.”

Page 2, line 63, revise “Wang L et al.” to “Wang et al.”.

Page 2, line 67-69, The statement, “According to preliminary studies, we found the volatile oils of SD had antioxidant and antitumor activity [20-21] and hoped to explore and find target molecules in the volatiles on this basis.”, is not clear. What does target molecules mean?

Page 2, line 69-74, It is not clear about the reason for using HS-SPME-GC/MS for SD analysis.  What is the benefit of using headspace SPME-GC/MS for SD analysis? What is the hypothesis in this work? It is not clear from the Introduction about multivariate statistical methods (HCA, PCA, and PLS-DA). Why these methods were used? What benefit the did authors expect from the use of these statistical methods? How these methods can help in answering the research questions of this research work?

Page 2, line 83, it is not clear how peaks in GC are defined? Were the peak areas integrated from TICs? How a peak was determined in the chromatogram?

Page 2, line 84, revise “Figure. 1-A” to “Figure 1-A”.

Page 3, line 93, revise “Figure.1-B” to “Figure 1-B”. The same issue with other figures should be revised in all the text.

Page 8, line 219, “Therefore, cis-anethol, cis-anethol, β-asarone and fitone were selected to analyze fragment information and identify the structure in the second-order mass spectrum.” What is the second-order mass spectrum? Only Agilent Technologies 6890N GC system coupled with an Agilent Technologies 5731 mass spectrometer has been described in section 3.4.

Page 8, line 220, did the author used GC/MS or GC/MS/MS? Why ESI-MS/MS is listed in Table 2?

Author Response

Dear Reviewer:

   We do extremely appreciate your suggestions. Based on them, we have made careful investigation, revision and improvement for the original manuscript. Main changes in the revised manuscript have been marked as highlighted text. We hope the revised manuscript will meet your requirement. The following content is our point-by-point responses to your comments and questions.

Looking forward to hearing from you soon, thank you very much!

Yours Sincerely

Authors

Response to reviewer

Comment 1: *P1/36: it is not clear which chemical component in SD has been used for cancer treatment.

Response: According to the following reference, some flavonoids such as amentoflavone, robustaflavone and heveaflavone were used for cancer treatment. The related contents could be seen in this paragraph. Thanks!

Lee et al. Amentoflavone inhibits ERK-modulated tumor progression in hepatocellular carcinoma in vitro. In vivo. 2018, 32, 549-562.

Comment 2: P2/44: authors state “In recent years, volatiles from SD have attracted wide attention [6]”, however, the cited reference was published in 1994, which was more than 20 years ago.

Response: Sorry, we have rechecked the reference and replaced it with the recent reference.

Comment 3: P2/44-46: authors state “The combination of headspace solid-phase microextraction and gas chromatography-mass spectrometry (HS-SPME/GC-MS) is an important method to study volatiles [7]”. The cited reference was not an HS-SPME/GC-MS work. Instead, it was microwave-assisted extraction work.

Response: Thank you for your advice. The wrong reference has been replaced with the correct one.

Comment 4: P2/51-52: some relevant literature below can be included in this paragraph:

Zakir Hossain, Barbara Bojko, Janusz Pawliszyn, Automated SPME–GC–MS monitoring of headspace metabolomic responses of E. coli to biologically active components extracted by the coating, Analytica Chimica Acta, Volume 776, 7 May 2013, Pages 41-49. Austin McDaniel, Lauren Perry, Qingzhong Liu, Wei-Chuan Shih, Jorn Yu, Toward the identification of marijuana varieties by headspace chemical forensics, Forensic Chemistry, Volume 11, December 2018, Pages 23-31.

Response: We have carefully read these literatures and cited them in our manuscript.

Comment 5: P2/53-53: a typical SPME does not use the membrane adsorption technology. Can the author verify the statement here? The authors should cite the original design of the SPME from “J. Pawliszyn, Solid Phase Microextraction: Theory and Practice, Wiley-VCH, New York, 1997.”

Response: We agree with the reviewer’s comments, and the original literature was cited. The verified statement could be seen in the revised manuscript.

Comment 6: P2/60: revise “Nie Cong-Ning et al” to “Nie et al.”

Response: This was our mistake. We have changed “Nie Cong-Ning et al” to “Nie et al”.

Comment 7: P2/63: revise “Wang L et al.” to “Wang et al.”.

Response: We have changed “Wang L et al” to “Wang et al”.

Comment 8: P2/67-69: The statement, “According to preliminary studies, we found the volatile oils of SD had antioxidant and antitumor activity [20-21] and hoped to explore and find target molecules in the volatiles on this basis.”, is not clear. What does target molecules mean?

Response: Thank you for this suggestion. As the reviewer pointed out, the words of “target molecules” are not appropriate. Actually, this work aims to find more chemical components in the volatiles of SD. We have corrected the mistake.

Comment 9:P2/69-74: What is the benefit of using headspace SPME-GC/MS for SD analysis?

Response: HS-SPME-GC/MS has not been used to analyze SD from different habitats before, and the author tried to explore the applicability of this method. The benefits of HS-SPME-GC/MS for SD analysis includes: (1) the technique integrates sampling, extraction and analysis, and thus it is easy to operate and shortens the analysis time; (2) it has the advantages of low sample consumption and high sensitivity. The related contents have been added in the second paragraph of Introduction. Thanks!

Comment 10: P2/69-74: What is the hypothesis in this work?

Response: The hypothesis in this work is to combine HS-SPME-GC/MS and multivariate analysis to discriminate SD from different habitats. The authors have proved that the approach is effective. Thanks.

Comment 11: P2/69-74: It is not clear from the Introduction about multivariate statistical methods (HCA, PCA, and PLS-DA). Why these methods were used?

Response: Multivariate statistical methods can be used to classify herbal medicines and find the variables for classification. HCA, PCA, and PLS-DA are multivariate statistical methods for classification from different perspectives. The authors tried to classify SD from multiple angles, and thus these methods were used. The relevant contents could be seen in the revised manuscript. Thanks.

Comment 12: P2/69-74: What benefit did authors expect from the use of these statistical methods?

Response: The authors expect to classify SD from different habitats by the use of these statistical methods. We have added the description in the last paragraph of Introduction. The results also demonstrated that these methods were effective. Thanks for the comments.

Comment 13: P2/69-74: How these methods can help in answering the research questions of this research work?

Response: HS-SPME-GC/MS can provide information on the chemical components of SD, and multivariate statistical methods distinguish SD from different habitats based on the information. By combining these methods, the influence of environment on the chemical components of SD could be analyzed. The relevant contents could be seen in the revised manuscript. Thanks!

Comment 14: P2/83: it is not clear how peaks in GC are defined?

Response: The peaks in GC are defined by a method of peak area normalization as shown in section 3.3.

Comment 15: P2/83: Were the peak areas integrated from TICs?

Response: Yes, the peak areas were integrated from TICs of GC-MS.

Comment 16: P2/83: How a peak was determined in the chromatogram?

Response: The components of the volatile oil from Selaginella doederleinii were identified using National Institute of Standards and Technology (NIST) 14.0 Mass Spectra Database and Wiley National Institute of Standards and Technology (Wiley NIST) 14.0 Mass Spectra Database. We have added the description in section 3.3.

Comment 17: P2/84: revise “Figure. 1-A” to “Figure 1-A”.

Response: We have changed “Figure. 1-A” to “Figure 1-A”.

Comment 18: P3/93: revise “Figure.1-B” to “Figure 1-B”. The same issue with other figures should be revised in all the text.

Response: We have changed “Figure. 1-B” to “Figure 1-B”. The same issue with other figures have be revised throughout the manuscript.

Comment 19: P8/219: “Therefore, cis-anethol, cis-anethol, β-asarone and fitone were selected to analyze fragment information and identify the structure in the second-order mass spectrum.” What is the second-order mass spectrum? Only Agilent Technologies 6890N GC system coupled with an Agilent Technologies 5731 mass spectrometer has been described in section 3.4.

Response: As the reviewer pointed out, the second-order mass spectrum is an unprofessional expression. Actually, it refers to MS spectrum. Besides, more experimental details have been added in Section 2.4. Thanks.

Comment 20: P8/220: Did the author used GC/MS or GC/MS/MS? Why ESI-MS/MS is listed in Table 2?

Response: Sorry, This is a typo. We used GC/MS, and the information obtained by ESI-MS was summarized in Table 2.

Reviewer 2 Report

Comments to the Author:

This manuscript studies the volatile composition in Selaginella doederleinii  using headspace solid phase  microextraction-gas chromatography-mass  spectrometry. In general terms the paper is well presented, the objectives are clearly defined and the tables are adequate. On the other hand, the paper is original and interesting, and I recommend it to be published with some minor revision.

-Line 54: “volatile components and aroma components”. What is the difference between them?

-Line 60: “Cong et al”. Authors must write “Cong et al.”

-Discussion must be more extended, there is few references. Authors must discuss more their results.

-Line 94: “volatile oils”; “ Line 96: “essentials oils”. During the text it can be found different words related to aroma composition: aroma, volatile, volatile oils, essentials…please specified each group of components or unified them.

 Table 1: Meaning of “Vip”.

-Line 263: Please, more information about from the batches are.

-Line 303: Please include information about the HCA analysis.

Author Response

Dear Reviewer:

   We do extremely appreciate your suggestions. Based on them, we have made careful investigation, revision and improvement for the original manuscript. Main changes in the revised manuscript have been marked as highlighted text. We hope the revised manuscript will meet your requirement. The following content is our point-by-point responses to your comments and questions.

Looking forward to hearing from you soon, thank you very much!

Yours Sincerely

Authors

Response to reviewer

Comment 1: P2/54: “volatile components and aroma components”. What is the difference between them?

Response: Actually, there is on difference between volatile components and aroma components in our work. We have deleted the repeated expression of aroma components in the revised manuscript. Thanks.

Comment 2: P2/60: “Cong et al”. Authors must write “Cong et al.”

Response: We have changed “Nie Cong-Ning et al” to “Nie et al”.

Comment 3: -Discussion must be more extended, there is few references. Authors must discuss more their results.

Response: As the reviewer suggested, we have extended discussion on the results. The references also increased from 15 to 42. Many thanks for all the valuable suggestions.

Comment 4: P3/94: “volatile oils”; P3/96: “essentials oils”. During the text it can be found different words related to aroma composition: aroma, volatile, volatile oils, essentials…please specified each group of components or unified them.

Response: As the reviewer’s suggestion, we have unified them throughout the manuscript.

Comment 5: Table 1: Meaning of “Vip”.

Response: Vip means variable important plot that reflects the importance of each evaluation index. The explanation of Vip could be seen in Table 1 and section 2.3.3. Thank you for the kind reminding.

Comment 6: P11/263: Please, more information about from the batches are.

Response: We have added more information about the batches of Selaginella doederleinii in Table 3. Thanks.

Comment 7: P12/303: Please include information about the HCA analysis.

Response: We have added the information about the HCA analysis in section 3.5.

Round 2

Reviewer 1 Report

Must be corrected before publication:

L170 Revision of Table 1 is needed. For example, some chemical formula has been separated into two lines. Also, the list of chemicals after No. 58 in Table 1 should be listed in a separate table. The list of chemicals is no line up well with the column names in Table 1.

L245 Table 2 According to Section 3.4, “An Agilent Technologies 6890N GC system coupled with an Agilent Technologies 5731 mass spectrometer was applied for component analysis.” Can this GC-MS perform ESI? It is very problematic to see “ESI-MS (m/z)” in Table 2. What is ESI? How to perform ESI  with Agilent GC/MS (what kind of ionization source has been installed in the 5731 MSD)?

Minor error:

L163 missing a period at the end of the sentence.

Author Response

Dear Reviewer:

   We are truly grateful to yours critical comments and thoughtful suggestions. Based on these comments and suggestions, we have made careful modifications on the original manuscript. All changes have been marked as highlighted text. We hope the new manuscript will meet your standard. Below you will find our point-by-point responses to your comments/ questions:

Looking forward to hearing from you soon, thank you very much!

Yours Sincerely

Authors

Response to reviewer

Comment 1: L167: Revision of Table 1 is needed. For example, some chemical formula has been separated into two lines.

Response: We have carefully rechecked and revised the errors in Table 1.

Comment 2: L170: the list of chemicals after No. 58 in Table 1 should be listed in a separate table. The list of chemicals is no line up well with the column names in Table 1.

Response: We have deleted this part content in Table 1 and added it in Table 2.

Comment 3:L245: According to Section 3.4, “An Agilent Technologies 6890N GC system coupled with an Agilent Technologies 5731 mass spectrometer was applied for component analysis.” Can this GC-MS perform ESI?

Response: Sorry, this is our mistake. GC-MS can perform EI and not ESI. But this MS has low resolution and the data analysis on protonated molecular ions is not reliable. We have deleted Section 2.4 in the manuscript. Thanks for the comments.

Comment 4:L317: It is very problematic to see “ESI-MS (m/z)” in Table 2. What is ESI?

Response: ESI is an electrospray ionization. It is mainly used in LC-MS. However, EI (Electron Ionization) is mainly used for ionization of volatile samples in GC-MS. This is a mistake in Table 2 and we have removed it.

Comment 5:L317: How to perform ESI with Agilent GC/MS (what kind of ionization source has been installed in the 5731 MSD)?

Response: Sorry, EI has been installed in the 5973MSD. Agilent GC/MS can perform EI and not ESI. The advantages of EI includes stable operation, rich fragmentation structural information, and retrieved standard mass spectra. Thanks for the comments.

Comment 6:L163: L163 missing a period at the end of the sentence.

Response: We have corrected this error in the revised manuscript.